# Hot Carrier Stress Sensing Bulk Current for 28 nm Stacked High-k nMOSFETs

**Chii-Wen Chen [1], Mu-Chun Wang [1],*, Cheng-Hsun-Tony Chang [1], Wei-Lun Chu [1], Shun-Ping Sung [2] and Wen-How Lan [3],***

[1] Department of Electronic Engineering, Minghsin University of Science and Technology, Hsinchu 30401, Taiwan; cwchen@must.edu.tw (C.-W.C.); chtchang@must.edu.tw (C.-H.-T.C.); vensonpig@gmail.com (W.-L.C.)

[2] Graduate Institute of Mechatronic Engineering, National Taipei University of Technology, Taipei 10608, Taiwan; sam1234808@gmail.com

[3] Department of Electrical Engineering, National University of Kaohsiung, Kaohsiung 81148, Taiwan

*   Correspondence: mucwang@must.edu.tw (M.-C.W.); whlan@nuk.edu.tw (W.-H.L.); Tel.: +886-3-559-3142 (M.-C.W.)

**Abstract:** This work primarily focuses on the degradation degree of bulk current ($I_B$) for 28 nm stacked high-k (HK) n-channel metal–oxide–semiconductor field-effect transistors (MOSFETs), sensed and stressed with the channel-hot-carrier test and the drain-avalanche-hot-carrier test, and uses a lifetime model to extract the lifetime of the tested devices. The results show that when $I_B$ reaches its maximum, the ratio of $V_{GS}/V_{DS}$ values at this point, in the meanwhile, gradually increases in the tested devices from the long-channel to the short ones, not just located at one-third to one half. The possible ratiocination is due to the ON-current ($I_{DS}$), in which the short-channel devices provide larger $I_{DS}$ impacting the drain junction and generating more hole carriers at the surface channel near the drain site. In addition, the decrease in $I_B$ after hot-carrier stress is not only the increment in threshold voltage $V_T$ inducing the decrease in $I_{DS}$, but also the increment in the recombination rate due to the mechanism of diffusion current. Ultimately, the device lifetime uses Berkley's model to extract the slope parameter *m* of the lifetime model. Previous studies have reported *m*-values ranging from 2.9 to 3.3, but in this case, approximately 1.1. This possibly means that the critical energy of the generated interface state becomes smaller, as is the barrier height of the HK dielectric to the conventional silicon dioxide as the gate oxide.

**Keywords:** drain current; hot-carrier; impact ionization; high-k; lifetime; substrate current; ALD technology

## 1. Introduction

With the continuous scaling of complementary metal–oxide–semiconductor (CMOS) technology, there are many benefits to metal–oxide–semiconductor field-effect transistors (MOSFETs), including an increasing number of devices in integrated circuits, not only providing impressive electrical performance of the device, but also decreasing the entire power consumption. However, the shorter channel length and thinner gate dielectric thickness of the MOSFET increase the OFF-current including the gate leakage and source/drain (S/D) punch-through effect, and relatively influencing the threshold voltage ($V_T$), short channel effect, and reliability issues [1,2]. In terms of reliability, the device lifetime will be reduced by hot carrier injection (HCI) [3,4]. The most common physical model for HCI is the lucky

carry model built by Hu et al. [5]. Berkeley's model can derive the device lifetime with bulk current (or called substrate current ($I_{SUB}$)). The model is shown in Equation (1).

$$\tau \times I_{DS} \propto \left(\frac{I_{SUB}}{I_{DS}}\right)^{-m} \tag{1}$$

where $\tau$ is the device lifetime, $I_{DS}$ is the drain-to-source current, $I_{SUB}$ is the substrate current, and the acceleration factor $m = \phi_{it}/\phi_i$, where $\phi_{it}$ is the critical hot carrier energy required to create an interface state of approximately 3.7 eV and $\phi_i$ is the minimum hot carrier energy required to create an impact ionization of approximately 1.3 eV for the poly-gate and Si-SiO$_2$ interface.

The hot carrier effect can be classified into two types: Channel-hot-carrier (CHC) and drain-avalanche-hot carrier (DAHC) tests [6]. The CHC effect means that the carriers near the drain terminal are accelerated by the lateral electric field and travel through the channel [7–12], as shown in Figure 1. The quoted references related to hot-carrier (HC) effects are listed in Table 1. Studies have shown that the maximum $I_{SUB}$ ($I_{SUB\_max}$) is at $V_{DS} = V_{GS}$ [13,14]. As $V_{DS} > V_{GS}$, the depletion region near the drain site is increased. As the carriers in the channel travel through this region, they are accelerated and energized to become hot carriers. These hot carriers may generate extra electron–hole pairs [15] in the channel, especially in the depletion region of the drain size. This phenomenon is called impact ionization. The generated electron may inject into the gate or drain terminal, and the generated holes trend to the substrate, as shown in Figure 2. The results demonstrate that the $I_{SUB\_max}$ value is located at $V_{GS} = V_{DS}/3 \sim V_{DS}/2$ [16,17], called the DAHC effect.

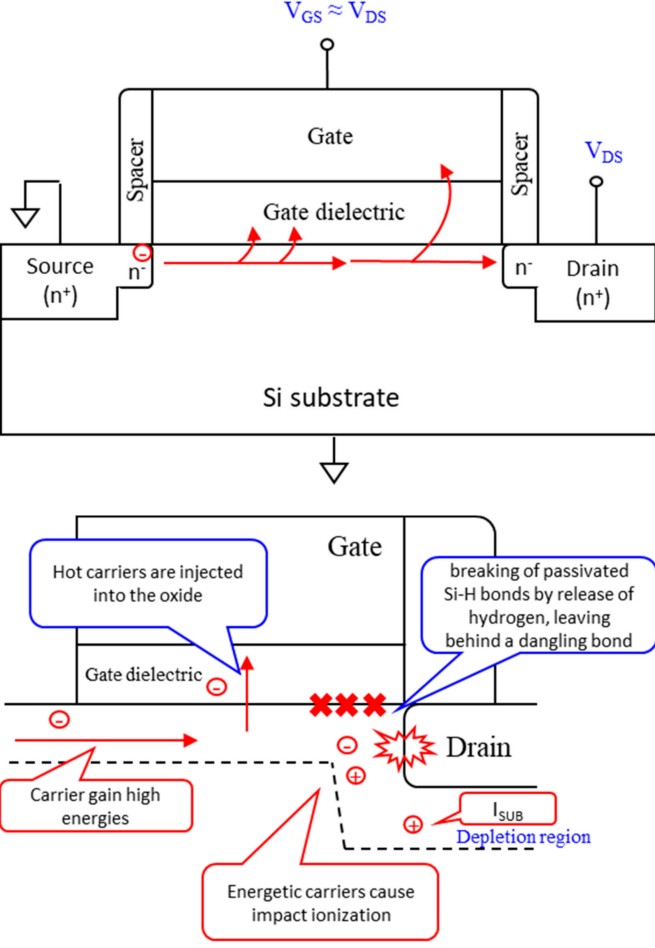

**Figure 1.** Schematic diagram of channel-hot-carrier injection.

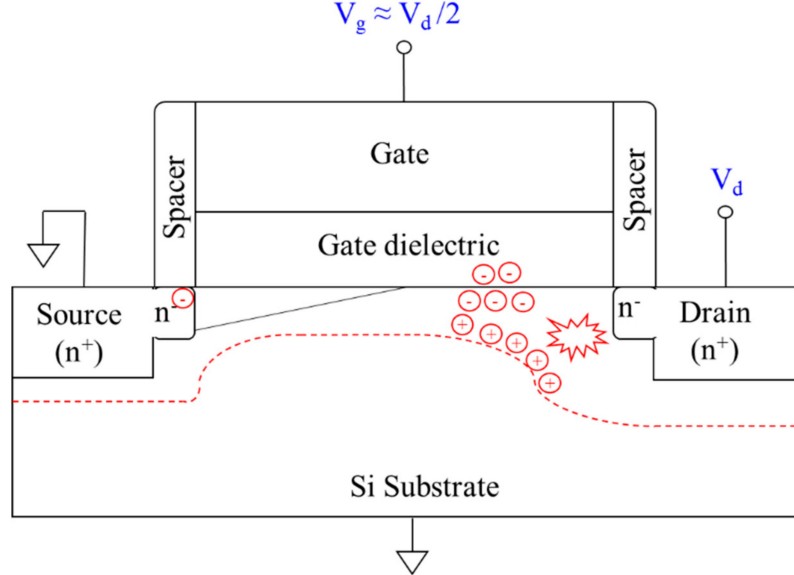

**Figure 2.** Major mechanism of drain-avalanche-hot-carrier generation.

In particular, while the thickness of the silicon dioxide ($SiO_2$)-based gate dielectric approaching its physical limitation, at the 45 nm-node-technology generation [18–20], is below 15 Å, gate leakage due to the direct tunneling effect cannot be tolerated at the OFF-current specification. Using the stacked high-k (HK) dielectric ($HfO_x$/$ZrO_y$/$HfO_z$ (HZH)), replacing the conventional $SiO_2$ in gate engineering or halo implants and lightly doped-drain (LDD) technology in the surface channel is a couple of attractive metrologies to promote the drive current and decrease the leakage in nano-node process technology [21].

**Table 1.** The comparison of quoted references [7–12,17,22] in the former and present process technologies.

| Reference | Purpose | Stress Method | Specifications |
|---|---|---|---|
| Takeda et al. [7] | Probing the DAHC injection and the substrate current-induced hot-electron injection (SCHE) under submicron process with gate dioxide. | DAHC and SCHE | • n-MOSFET with 0.8 µm process<br>• Gate dioxide, $T_{ox} \approx 10$ nm<br>• CADDET simulator<br>• DAHC dominating HCD |
| Abramo et al. [8] | Using a Monte Carlo simulator to quantify the electron energy distribution in Si devices at low applied voltages involving the process of carrier heating. | DAHC | • n-MOSFET with 0.25 µm process<br>• Gate dioxide, $T_{ox} \approx 5$ nm<br>• Monte Carlo Simulator<br>• Electron energy distribution<br>• Carrier heating<br>• Electron–electron interaction |
| Yu et al. [9] | Exposing the hot-carrier effect related to the channel implantation process influencing the normal and reverse short-channel effect LDD MOSFETs down to 0.1 µm. | CHC | • n-MOSFET with 0.1 µm process<br>• Gate dioxide, $T_{ox} \approx 2.6$ nm<br>• CADDET simulator<br>• reverse short-channel effect<br>• Pocket implant |
| La Rosa et al. [10] | Reviewing the CHC mechanism and its effects on n-MOSFETs of deep submicron CMOS bulk technologies guided into the carrier dominant energy. | CHC | • n-MOSFET below 0.25 µm process<br>• Gate dioxide, $T_{ox} \approx 3.3$ nm<br>• R-D model<br>• Forming gas, $H_2$<br>• Carrier dominant energy |

**Table 1.** *Cont.*

| Reference | Purpose | Stress Method | Specifications |
|---|---|---|---|
| Mahapatra et al. [11] | Reviewing the physical mechanisms of transistor parameter shift due to hot-carrier degradation (HCD) in n-MOSFETs. | CHC and DAHC | • Channel length: ~2 μm to ~20 nm<br>• Gate dioxide, $T_{ox}$: 20 to 1 nm<br>• $V_D$: 10 to 1 V<br>• S/D junction depths |
| Mahapatra et al. [12] | Reviewing the technology scaling including the stress temperature and performing the comparison of dc and ac stress | CHC under dc and ac stress | • Time kinetics<br>• N-MOSFET and FinFET<br>• LDD and SDE n-MOSFETs |
| Acovic et al. [17] | Reviewing the hot-carrier effects and reliability problem in MOSFET | DAHC and ac stress | • Time kinetics<br>• SOI MOSFET and bulk MOSFET<br>• Effects of scaling on the HCD<br>• Stress temperature effect |
| Amat et al. [22] | Presenting a comprehensive study on CHC degradation in short-channel MOSFETs with high-k dielectric | CHC and ac stress | • BTI effect<br>• Modified the LEM model.<br>• Quasi-static behavior |
| This work | Studying the degradation mechanisms of substrate current for high-k MOSFETs after HC stresses and exposing the change mechanisms of values of the acceleration factor in lifetime calculation | CHC and DAHC | • Lower Vcc (= 0.8 V)<br>• Lower barrier height<br>• High-k dielectric (EOT: ~22 Å)<br>• Diffusion current model for the nano channel length |

In this study, we used a HK-stack and metal gate (MG) as the n-MOSFET structure to analyze the variation in substrate current under hot carrier stresses [22]. In Equation (1), the severity of hot carrier injection is observed by the degradation in the substrate current, related to the issues of device lifetime. The other interesting event is to expose the relationship between channel lengths and substrate currents in nano-node n-MOSFETs. In addition, we used different stress conditions to probe the impact of substrate current, and investigated the $\tau \times I_{DS}/W$ vs. The $I_{SUB}/I_{DS}$ model extracts the slope parameter *m* in Equation (1). In this study, $V_T$ extraction was followed the constant current methodology. To accelerate the process and circuit development in yield and reliability analysis in the nano-node era, the technology computer-aided design simulator is an appropriate choice as an assistant [23–25].

## 2. Concise Process Flow and Stress Conditions

In this study, the schematic tested devices on 28 nm HK/MG wafers fabricated from the United Microelectronics Corporation (UMC) are used to perform the related extraction and analysis, as shown in Figure 3. After the standard cleaning, an interfacial layer (IL) of $SiO_x$ of approximately 9 Å was grown thermally to play a buffer between the surface channel and HK material and resist the nitrogen free radical to arrive at the surface channel to form the silicon nitride. Subsequently, the HK material was deposited as HZH by atomic layer deposition (ALD) technology [26–28]. In sequence, the devices were processed with the decoupled plasma nitridation (DPN) treatment to retard the amount of oxygen vacancies [29,30]. The treatment process employed the annealing temperature (700 °C) and nitrogen concentration (8%) after accomplishing an HK layer. The other key processes include an Si-based substrate, channel implantation, S/D engineering, interfacial layer, barrier metal, and low-resistivity Al metal gate. The metal gate was adopted in the gate-last (GL) process technology [31]. This technology provides several good functions to reduce the threshold voltage, gate electrode resistance, power consumption, and gate delay. The detailed 28 nm HK/MG process flow with the GL process can be referred to Wang et al. [21].

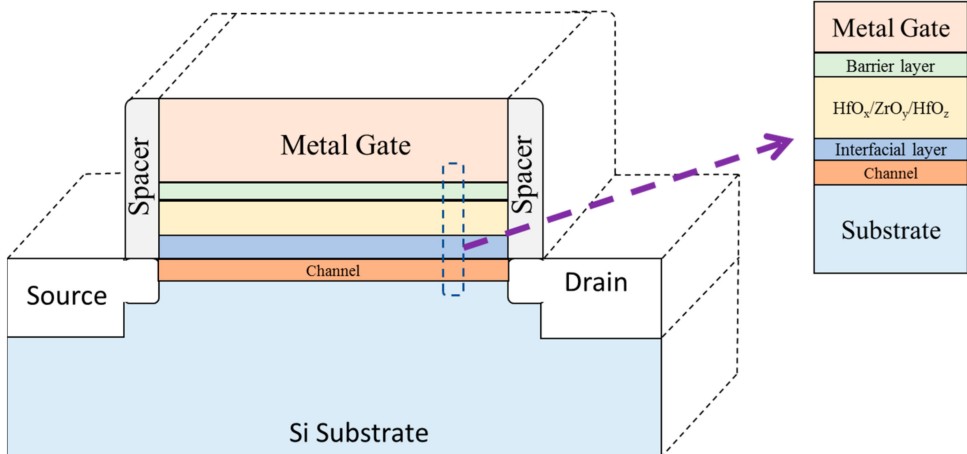

**Figure 3.** Schematic cross-section profile of an n-channel metal–oxide–semiconductor field-effect transistor (n-MOSFET).

The basic electrical characteristics and stress tests were performed using a Keithley 4200 Semiconductor Characterization System. The test conditions can be divided into two parts. The first part is that the $I_{SUB}$ is measured with different channel lengths to observe the impact of the channel length change. The measurement conditions are listed in Table 2. The second part is related to the measurement of the HC degradation test. In these stresses, different stress voltages and channel lengths were applied to sense and analyze the most serious stress method, either CHC or DAHC stress [32]. The measurement conditions are presented in Table 3 with the $V_T$ extraction metrology following the constant current measurement.

**Table 2.** Measurement conditions for $I_{SUB}$ values.

| Variable | Parameters Setup |
|---|---|
| Device width (μm) | 0.3 |
| Vcc (V) | 0.8 |
| Dimension (μm) | 0.027, 0.03, 0.04, 0.05, 0.06, 0.08, 0.1, 0.5, 1 |
| Stress conditions | $V_{GS} = -0.4$–2.3 V; $V_{DS} = 2.3$ V |
| Temperature (°C) | 25 |

**Table 3.** Test conditions under HC stresses.

| Variable | Parameters Setup | |
|---|---|---|
| Device width (μm) | 0.5 | |
| Vcc (V) | 0.8 | |
| Stress mode | CHC | DAHC |
| Dimension (μm) | 0.03, 0.07, 0.11 | |
| $V_{stress}$ | $V_{GS} = V_{DS} = 1.2$ V, 1.4 V, 1.6 V | $V_{GS}$ at $I_{SUB, max}$; $V_{DS} = 1.2$, 1.4, 1.6 V |
| Temperature (°C) | 25 | |
| Threshold Voltage ($V_T$) | Constant current method to extract $V_T$ values, $V_{T, lin} = V_{GS}$ at $I_{DS} = 300$ nA × W/L, $V_{DS} = 0.1$ V, $V_B = V_S = 0$ V | |

## 3. Results and Discussion

### 3.1. The Relationship between the Channel Length and $I_{SUB}$

The $I_{SUB}$–$V_{GS}$ curves of n-MOSFETs were measured for different channel lengths, as shown in Figure 4. When the tested device was in the OFF state, the increment in absolute value of gate voltage at the negative $V_{GS}$ axis increased the $I_{SUB}$ because of the gate-induced drain leakage effect [33]. However, as the tested device was operated in the ON state, the gate voltage increased, inducing an increment in $I_{SUB}$. This effect points to the fact that the average carriers in short channels are hotter and have

more energy to create an impact ionization event. Thus, the rate of increase in $I_{SUB}$ with the shorter channel-length device is higher than the increase in $I_{DS}$. As the channel length of the tested device decreases, the lateral electric field increased, and the $I_{DS}$ increases, resulting in an impact ionization rate ($I_{SUB}/I_S$), where $I_S$ is the current sensed at the source terminal, which increases, as shown in Figure 4a. While the channel length decreased, the maximum $I_{SUB}$ with the increase in $V_{GS}$ increases, generating a higher impact ionization rate, as shown in Figure 4b. The possible ratiocination indicates that the gate voltage increases, indicating a stronger vertical field, to attract more inversion electrons to recombine the holes in the longer channel length.

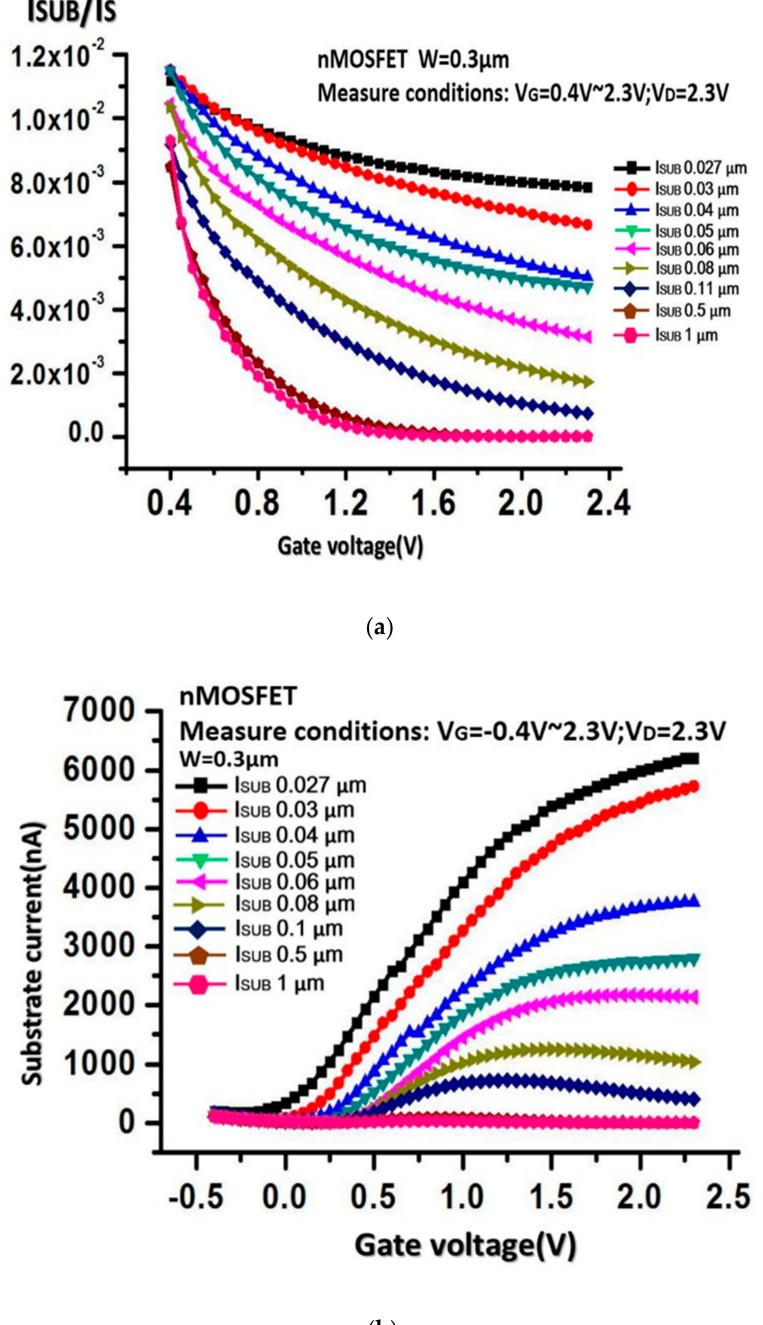

(**a**)

(**b**)

**Figure 4.** (**a**) The impact ionization compared with different channel lengths; (**b**) substrate current with different channel lengths of n-MOSFETs.

### 3.2. $I_{SUB}$ and $V_T$ Degradation after HC Stresses

The stress conditions in the CHC stress mode are listed in Table 3. Figure 5 illustrates that the short channel device causes serious degradation owing to the larger horizontal field from the drain site to the source node, and the higher stress voltage can also increase the device degradation in the $I_{SUB}$ aspect. In summary, the most obvious degradation for $I_{SUB}$ is set at the tested device $W/L = 0.5/0.03$ μm and the stress voltage at $V_{GS} = V_{DS} = 1.6$ V. The other degradation index $V_T$ shifts with different channel lengths, and the different stress voltages are shown in Figure 6. The $V_T$ shift observed at $L = 0.03$ μm and the stress voltage $V_{GS} = V_{DS} = 1.6$ V are the worst [34–36]. However, as the stress condition is at the higher gate field, the distribution of $V_T$ shift at $L = 0.03$ μm and 0.11 μm is similar, as shown in Figure 6b,d, but not at the lower field, as shown in Figure 6a,c. This speculation is that the capability of trap repair in the lower field is better in the channel, and so is the longer device. Following the research results of Huang et al. [37], they denoted that the drain current in the nano-MOSFETs covers the drift and diffusion currents, fitting well in simulation and measured electrical current–voltage characteristics. In this reference, the carrier conduction in the channel is similar to that in a *p-n* junction. The entire current flow in the *p-n* junction is mainly dominated by the diffusion mechanism. Therefore, a reasonable speculation of the decrement in $I_{SUB}$ after HC stress is not only the increase in $V_T$ indirectly causing the decrease in $I_{DS}$, but also the increase in the recombination rate arising from the diffusion current, especially for the nano-node devices. As the channel length is less than 0.04 μm, this consequence is more distinct regardless of the HC stress method.

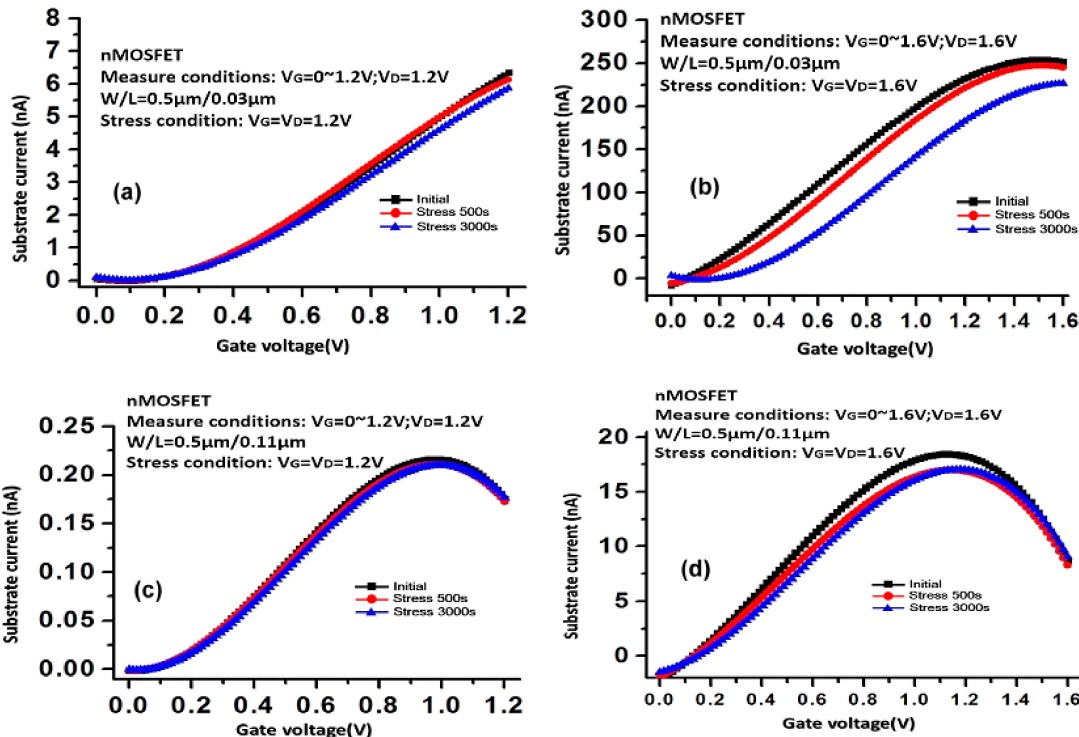

**Figure 5.** In channel-hot-carrier (CHC) test conditions, the $I_{SUB}$ at: (**a**) $W/L = 0.5/0.03$ μm/μm, stressed at $V_{GS} = V_{DS} = 1.2$ V, (**b**) $W/L = 0.5/0.03$ μm/μm, stressed at $V_{GS} = V_{DS} = 1.6$ V, (**c**) $W/L = 0.5/0.11$ μm/μm, stressed at $V_{GS} = V_{DS} = 1.2$ V, and (**d**) $W/L = 0.5/0.11$ μm/μm, stressed at $V_{GS} = V_{DS} = 1.6$ V.

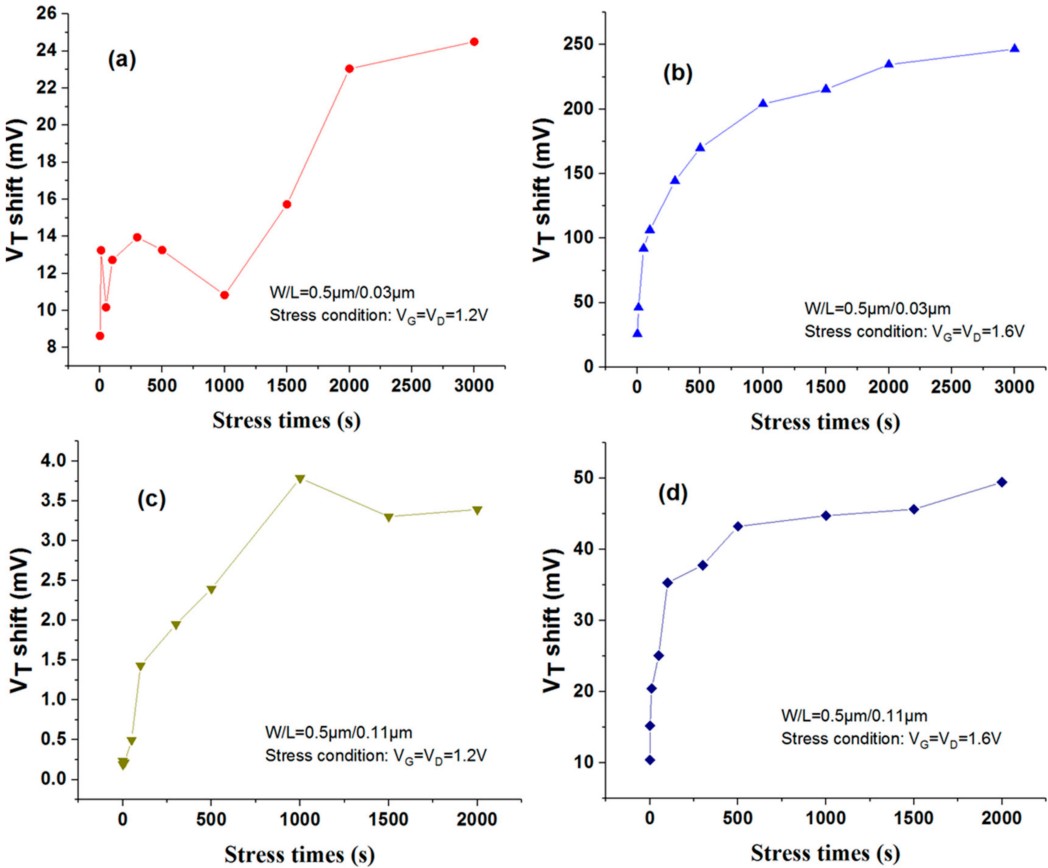

**Figure 6.** In CHC test conditions, the $V_T$ shift at: (**a**) $W/L = 0.5/0.03$ µm/µm, stressed at $V_{GS} = V_{DS} = 1.2$ V, (**b**) $W/L = 0.5/0.03$ µm/µm, stressed at $V_{GS} = V_{DS} = 1.6$ V, (**c**) $W/L = 0.5/0.11$ µm/µm, stressed at $V_{GS} = V_{DS} = 1.2$ V, and (**d**) $W/L = 0.5/0.11$ µm/µm, stressed at $V_{GS} = V_{DS} = 1.6$ V.

In the DAHC stress mode, the stress conditions are similar to the CHC stress. The slight difference is that the gate voltage is defined by the $V_{GS}$ at the maximum $I_{SUB}$. The test results show that the maximum $I_{SUB}$ does not appear in the short-channel device ($L = 0.03$ µm). Therefore, the discussion will only focus on the tested devices with $L = 0.07$ µm and $L = 0.11$ µm. Figure 7 shows that the most serious degradation of $I_{SUB}$ is at $L = 0.07$ µm and the stress voltage $V_{DS} = 1.6$ V. The $V_T$ shift is shown in Figure 8. The worst degradation is at $L = 0.07$ µm and the stress voltage $V_{GS}$ at $I_{SUB\_max}$ and $V_{DS} = 1.6$ V. After the DAHC stress, the substrate current is similarly decreased by the increase in $V_T$, causing a decrease in $I_{DS}$ and an increase in the recombination rate from the diffusion current.

By observing the amount of $V_T$ shift with CHC and DAHC tests, the value of $V_T$-shift with the CHC test is higher than that with the DAHC test. This phenomenon is similar to that reported in [32]. As deep analysis, because of the low gate field, the distribution trends of $V_T$ shift vs. stress time are not the same, which are different from the consequences under the CHC stress. However, the decrease trends of substrate current both before and after the HC stresses seem compatible.

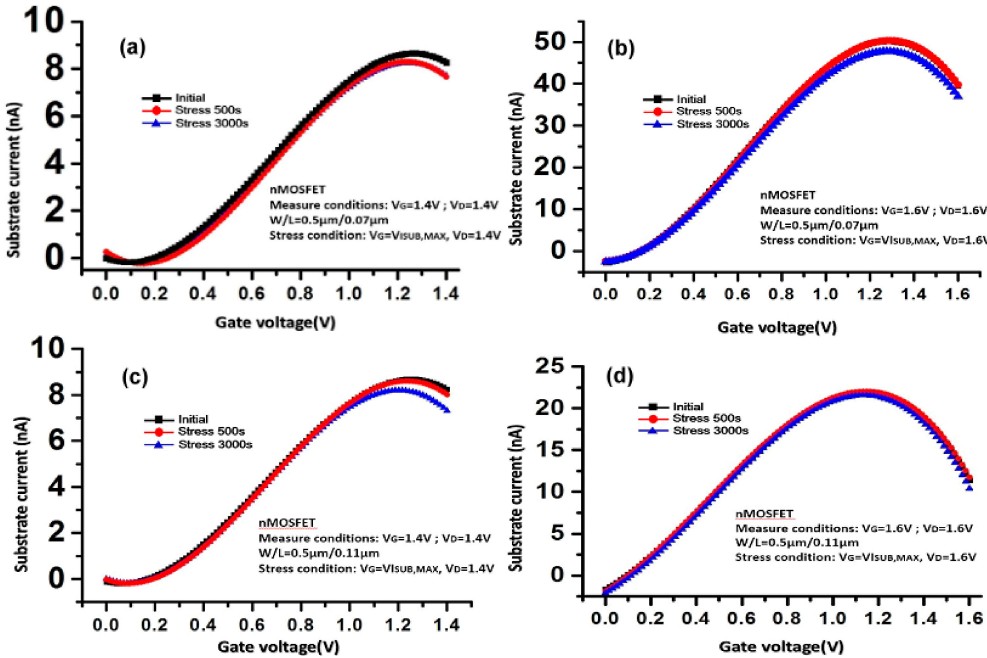

**Figure 7.** In drain-avalanche-hot carrier (DAHC) test mode, the $I_{SUB}$ at: (**a**) $W/L$ = 0.5/0.07 μm/μm, stressed at $V_{GS}$ at $I_{SUB\_max}$ plus $V_{DS}$ = 1.4 V, (**b**) $W/L$ = 0.5/0.07μm/μm, stressed at $V_{GS}$ at $I_{SUB\_max}$ and $V_{DS}$ = 1.6 V, (**c**) $W/L$ = 0.5/0.11 μm/μm, stressed at $V_{GS}$ at $I_{SUB\_max}$ and $V_{DS}$ = 1.4 V, and (**d**) $W/L$ = 0.5/0.11 μm/μm, stressed at $V_{GS}$ at $I_{SUB\_max}$ and $V_{DS}$ = 1.6 V.

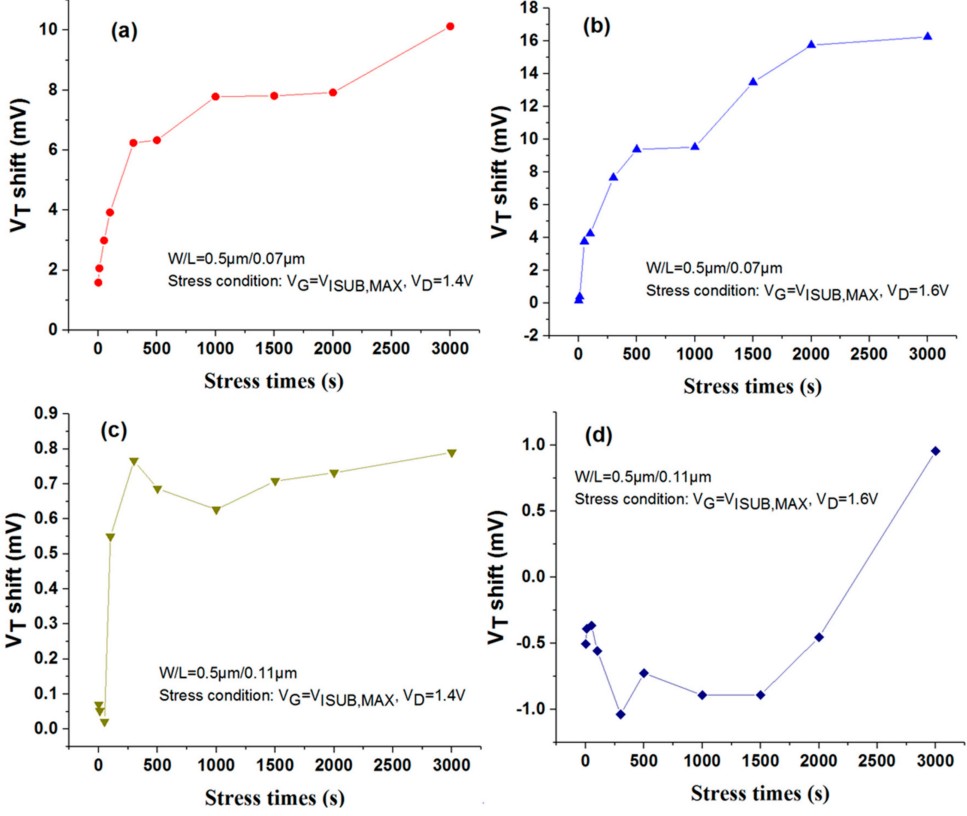

**Figure 8.** In DAHC test mode, the $V_T$ shift at: (**a**) $W/L$ = 0.5/0.07 μm/μm, stressed at $V_{GS}$ at $I_{SUB\_max}$ plus $V_{DS}$ = 1.4 V, (**b**) $W/L$ = 0.5/0.07 μm/μm, stressed at $V_{GS}$ at $I_{SUB\_max}$ and $V_{DS}$ = 1.6 V, (**c**) $W/L$ = 0.5/0.11 μm/μm, stressed at $V_{GS}$ at $I_{SUB\_MAX}$ and $V_{DS}$ = 1.4 V, and (**d**) $W/L$ = 0.5/0.11 μm/μm, stressed at $V_{GS}$ @$I_{SUB\_max}$ and $V_{DS}$ = 1.6 V.

### 3.3. HC Lifetime Model for n-MOSFETs

The $\tau \times I_{DS}/W$ vs. $I_{SUB}/I_{DS}$ model [5,16] is adopted into these tested devices, as in Equation (1). The model can effectively explain the correlation between $I_{SUB}$ and lifetime ($\tau$). When the $I_{SUB}$ increases, the lifetime decreases. This indicates that when the $I_{SUB}$ is larger, the HC effect and the device degradation become serious. The slope of the predicted line, *m*, is 1.1, as shown in Figure 9, compared with the former research *m* ranging from 2.9 to 3.3 for SiO$_2$. The decrement in *m*-value means that the interface states become easier to generate due to the HK structure. According to references [38,39], the work function of polygrain Al is 4.13 eV and the band offset of HfO$_2$ compared to Si is 1.5 eV. The affinity of Si is 4.05 V. The barrier height between gate Al and HfO$_2$ is approximately 1.58 eV, close to the critical hot carrier energy, requiring the creation of an interface state $\phi_{it}$, as shown in Figure 10. If we adopt the $\phi_i$ = 1.3 eV minimum hot carrier energy to create an impact ionization in the Si-based surface channel, the ratio of $\phi_{it}/\phi_i$ representing the theoretical *m*-value is approximately 1.2, which is very close to the extracted m parameter 1.1.

After stress, the subthreshold swing *SS* is changed, related to the change in interface integrity between the IL and Si-based channel. The $\Delta SS$ (*SS* value after stress—*SS* value before stress) is equal to

$$\Delta SS = 2.3 \frac{kT}{q} \cdot \frac{\Delta C_{it}}{C_{ox}} \qquad (2)$$

where *k* is Boltzmann's constant, *T* is the absolute temperature, q is the unit charge, $C_{it}$ is the equivalent interface-state capacitance per area = q$D_{it}$, $D_{it}$ is the interface state density, $N_{it}$ is the interface state number per area with integration of $D_{it}$ in the energy band, and $C_{ox}$ is the gate capacitance per area.

The threshold voltage change, $\Delta V_T$, after stress contains the change in the oxide trap in the gate dielectric and the interface state on the surface channel, as shown in Equation (3). $\Delta Q_f$ covers the q$N_{it}$ change q$\Delta N_{it}$ and q$N_{ot}$ change q$\Delta N_{ot}$, where $N_{ot}$ is the oxide trap number per area in the gate dielectric.

$$\Delta V_T = \frac{\Delta Q_f}{C_{ox}} \qquad (3)$$

Using Equations (2) and (3), $\Delta N_{it}$ and $\Delta N_{ot}$ can be decoupled after hot carrier stress. These two amounts also explain the degradation level of the oxide trap and interface trap state for a tested device under a long-time operation, as shown in Figure 11 with *W/L* = 1/0.03 µm under different plasma nitridation treatments [21,39,40]. $\Delta N_{it}$ or $\Delta N_{ot}$ with different nitridation treatments exposes the different historical trends in hot carrier stress. In addition, in terms of the test consequences, the $V_T$-shift with CHC stress is more serious than that with DAHC, as shown in Figure 12. Even though the observed $I_{SUB\_max}$ occurs well under DAHC stress conditions, the major degradation mechanism still comes from the interface state and/or oxide trap generation [41–44]. Moreover, the generation of the interface state near the IL is also possibly due to the channels strained, which could be more relevant to reduce the bonding energy than that on the top of the HK layer. Due to the CHC stress mode owing to the higher gate voltage generating more interface states and oxide traps, the $V_T$-shift in the worst case under the view of $I_{SUB\_max}$ can be effectively demonstrated to be attributed to the $V_{GS} = V_{DS}$ stress condition, not at $V_{GS}$ traditionally located at one-third to one-half $V_{DS}$ [5].

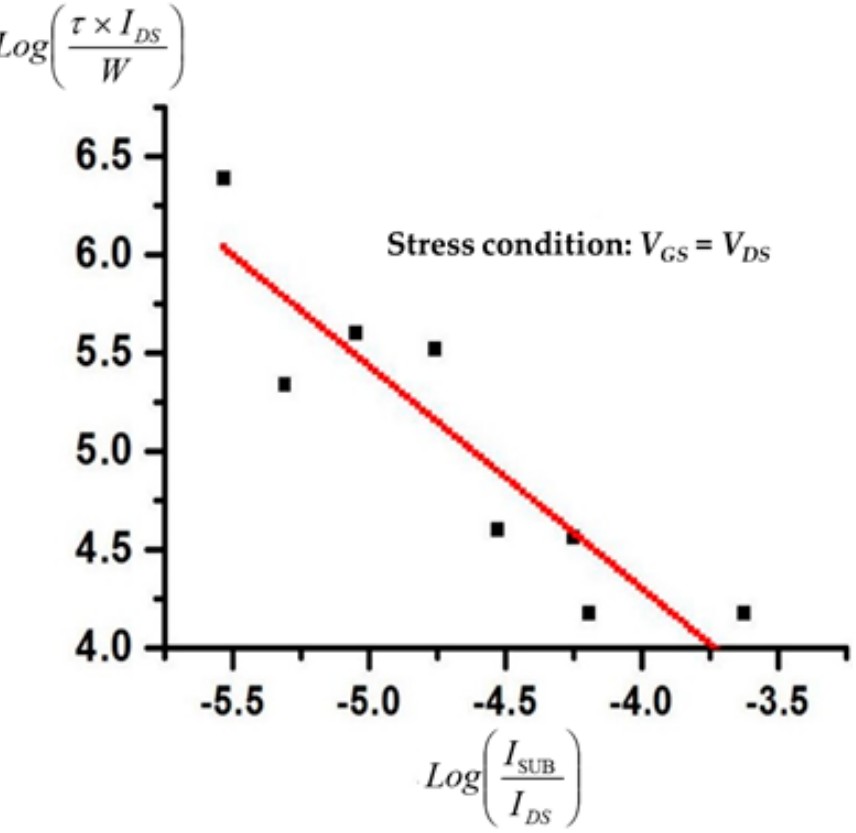

**Figure 9.** n-MOSFETs lifetime data using the proposed model plotted on log ($\tau \times I_{DS}/W$) versus $I_{SUB}/I_{DS}$.

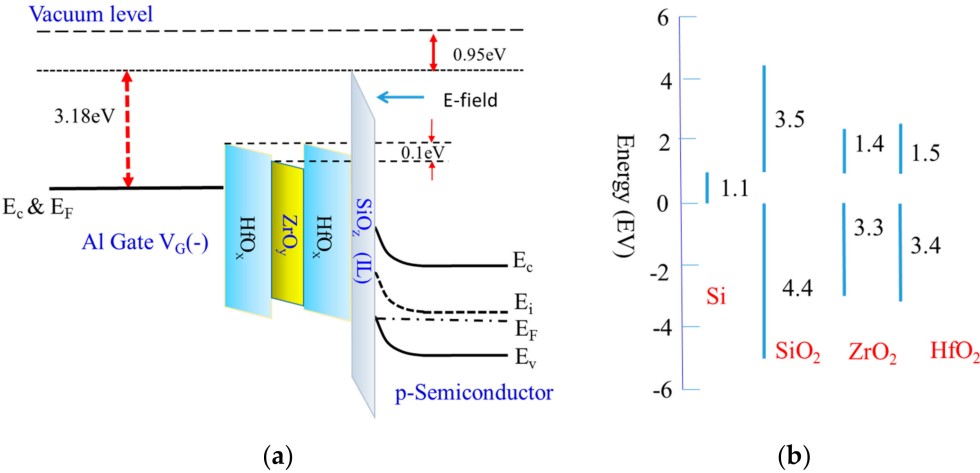

**Figure 10.** (**a**) Simple band diagram of high-k (HK)-stacked HfO$_x$/ZrO$_y$/HfO$_z$ (HZH) structure operated at accumulation mode. E$_C$: Conduction band, E$_i$: Intrinsic Fermi level, $E_F$: Fermi level, and E$_V$: Valence band, and (**b**) band offset for different dielectric compared with Si.

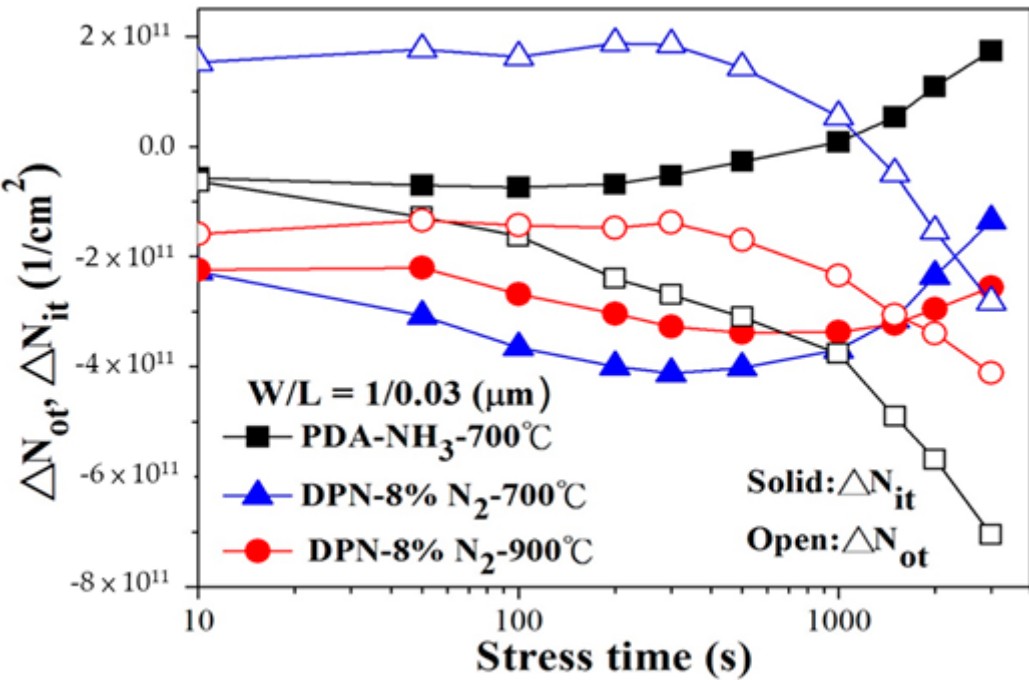

**Figure 11.** The shift in interface trapped charge and oxide trapped charge after 3000 s CHC stress ($W/L$ = 1/0.03 μm/μm) with decoupled plasma nitridation (DPN) or post-deposition annealing (PDA) treatments under 8% $N_2$ concentration and annealing temperatures.

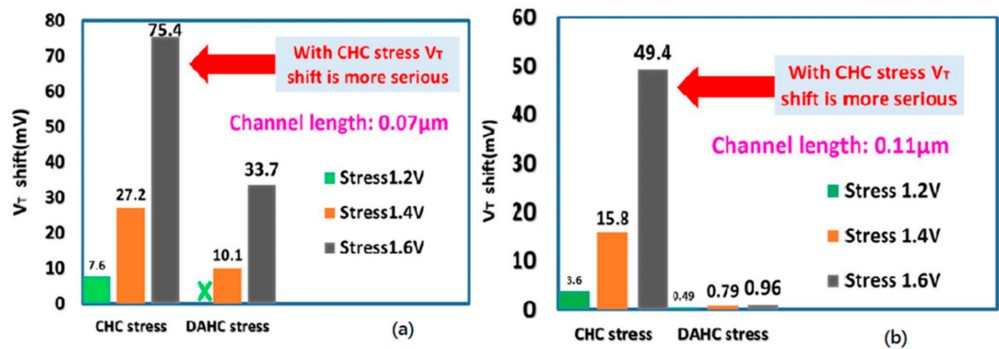

**Figure 12.** The $V_T$ shift compared the CHC test with the DAHC test as the tested devices: (**a**) $W/L$ = 0.5/0.07 μm/μm and (**b**) 0.5/0.11 μm/μm.

## 4. Conclusions

In this work, we observe that the maximum substrate current and the ratio of $I_{SUB}/I_s$ before the HC stress increased as the channel length of the tested devices was shorter. After the HC stress, the $I_{SUB\_max}$ decreased, especially for the deep nano-node channel-length device because of the increase in $V_T$ indirectly degrading the $I_{DS}$ and the increase in the recombination rate from the diffusion current as the channel length entered the nano-node level. Through the longer stress time, this phenomenon was more obvious, which also contributed to the $V_T$ shift. Even though the observed $I_{SUB\_max}$ occurred well under DAHC stress conditions, the major degradation mechanism still came from the interface state and/or oxide trap generation. The other consistent agreement was to extract the HC lifetime with Berkley's model, still available in HK/MG n-MOSFETs, deposited with ALD technology, but the values of acceleration factor $m$ were different from the gate dioxide or oxy-nitride. Ultimately, the HC stress is indeed and still a good gauge or application in nano-node device reliability tests or process splits in the optimal adjustment of front-end processes, such as channel implementation, growth of the gate dielectric, or HK dielectric deposition with nitridation treatment.

**Author Contributions:** Conceptualization, C.-W.C.; methodology, W.-H.L. and S.-P.S.; formal analysis, C.-W.C. and M.-C.W.; data curation, S.-P.S. and W.-L.C.; writing—Original draft preparation, M.W.; writing—Review and editing, M.-C.W.; project administration, C.-H.-T.C. All authors have read and agreed to the published version of the manuscript.

**Funding:** This research received no external funding.

**Acknowledgments:** The authors cordially thank United Microelectronics Corporation in Taiwan for providing precious 12″ wafers, and the financial support from Ministry of Science and Technology of Republic of China under Contract Nos. MOST 109-2622-E-159-001.

**Conflicts of Interest:** The authors declare no conflict of interest.

## Abbreviations

| | |
|---|---|
| $T_{ox}$ | Thickness of oxide |
| CADDET | Computer-Aided Device Design in Two Dimensions |
| R-D | Recombination-Diffusion |
| SDE | Source/Drain Extension |
| LEM | Lucky Electron Model |
| BTI | Bias Temperature Instability |
| EOT | Equivalent Oxide Thickness |

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
