# Peer review of "Hot Carrier Stress Sensing Bulk Current for 28 nm Stacked High-k nMOSFETs"

_electronics, doi:10.3390/electronics9122095_

Round 1

Reviewer 1 Report

This paper investigates hot carrier stress sensing bulk current for 28nm stacked high-k nMOSFETs. Case studies have also been performed. This is a well-organized and well-illustrated paper. I think that it deserves to be published after minor revisions. Here, there are some concerns of this reviewer:

1 The main contributions and novelty of this paper should be further summarized and clearly demonstrated. This reviewer suggests the authors exactly mention what is new compared with existing approaches and why the proposed approach is needed to be used instead of the existing methods.

2 Different figures can be made clearer to improve the readability of this paper.

3 The nomenclature should be included to help the reader to follow the paper conveniently.

Author Response

This paper investigates hot carrier stress sensing bulk current for 28nm stacked high-k nMOSFETs. Case studies have also been performed. This is a well-organized and well-illustrated paper. I think that it deserves to be published after minor revisions. Here, there are some concerns of this reviewer:

1 The main contributions and novelty of this paper should be further summarized and clearly demonstrated. This reviewer suggests the authors exactly mention what is new compared with existing approaches and why the proposed approach is needed to be used instead of the existing methods.

2 Different figures can be made clearer to improve the readability of this paper.

3 The nomenclature should be included to help the reader to follow the paper conveniently.

Reply:

        We really appreciate you for giving us these precious suggestions. We will follow them:

  1. Thanks for your notice. The trend shift of bulk current under different HC stress methods and channel-length device is further summarized and clearly demonstrated, especially in abstract and conclusion parts. In the past, no one focused on this trend shift. The difference of the trend shift of bulk current after HC stress between the traditional gate oxide (SiO2) and the high-k gate dielectric is strengthened in explanation more. The speculation of change of acceleration factor m in lifetime calculation is also demonstrated.
  2. The figure quality is adjusted more to fit the readability of this paper. Thanks for your great suggestion.

    3.Fine! The abbreviation of professional nomenclature will be introduced at the first time with the full-name expression.

Reviewer 2 Report

This study focuses on the degradation degree of bulk current (IB) for 28nm n15 channel MOSFETs, sensed and stressed with the channel-hot-carrier (CHC) test and the drain avalanche-hot-carrier (DAHC) test, and uses a lifetime model to extract the lifetime of tested devices. I have the following concerns:

  1. The article provides neither any new ideas nor even any contribution to the existing knowledge. 
  2. The authors should present more clearly the research gap through conducting a concise state-of-the-art analysis in the introduction section.
  3. Conclusions must go deeper, it would be more interesting if the authors focus more on the significance of their findings regarding the importance of the study.
  4. The literature review is very poor with backdated references.  Summarise all the literature as a table.
  5. Please avoid putting lump sum reference, such as [7--10]. Explain each reference.
  6. I will recommend seeing the format of a high impact journal. The way you presented is not the right way to write a journal paper.
  7. The writing quality should be improved.

Author Response

This study focuses on the degradation degree of bulk current (IB) for 28nm n-channel MOSFETs, sensed and stressed with the channel-hot-carrier (CHC) test and the drain avalanche-hot-carrier (DAHC) test, and uses a lifetime model to extract the lifetime of tested devices. I have the following concerns:

  1. The article provides neither any new ideas nor even any contribution to the existing knowledge.

The authors should present more clearly the research gap through conducting a concise state-of-the-art analysis in the introduction section.

  1. Conclusions must go deeper. It would be more interesting if the authors focus more on the significance of their findings regarding the importance of the study.
  2. The literature review is very poor with backdated references. Summarize all the literature as a table. Please avoid putting lump sum reference, such as [7-10]. Explain each reference.
  3. I will recommend seeing the format of a high impact journal. The way you presented is not the right way to write a journal paper. The writing quality should be improved.

Reply:

        Thanks for giving us the great illumination. We will follow your suggestion to amend them.

  1. In this article, we chiefly focus on the trend shift of the bulk current (or called substrate current) for the tested devices deposited with high-k gate dielectric before and after HC stress. And the ratio of ISUB/IS, which IS is the sensed current at the source terminal of the tested device, is increased as the channel length of the tested devices is decreased. After HC stress, the decrease of bulk current is not only the increase of VT indirectly causing the decrease of IDS, but the increase of recombination rate due to the diffusion mechanism, which is more obvious as the channel length is 0.03 μm, referring the updated Reference 36. Furthermore, these test consequences not only illustrate the extension of Berkley’s model in high-k gate dielectric, but the reduction of the acceleration factor m. The worst case of device degradation after HC stress with CHC or DAHC stress method is dedicatedly illustrated, which is still referable to the readers or researchers.
  2. Thanks for your notice. According to your advice, we strengthen the conclusion part and go deeper in the relation of this work. We add the possible mechanisms of decrease of the bulk current before and after HC stress more.
  3. Following your suggestion and the Reviewer #3’s comments, two hot-carrier subjects published in 2020 are included in the reference. The growth of high-k gate dielectric with nitridation treatment published in 2020 is also referred in this article. The backdated and novel references are demonstrated together to link the past and the advanced development with hot-carrier stress. The lump sum reference is allowed in the MDPI system such as electronics, biosensors, micromachines, and sensors. For instance, the electronics-template in Line 35 shows [4-6]. In this journal, electronics, the recent published paper in April 2020, doi: 10.3390/electronics9050712, shows [1-6] in the Line 33 of the Introduction part. Thus, our writing format should be allowable. However, due to your concerns, we list the illustration of Reference 7-12 as a Table, especially.
  4. After amending the previous three concerns, we send this manuscript to the proofreading service. The service website is “www.editage.com”, which provides the native English proofreading.

Reviewer 3 Report

The paper deals with the investigation of hot carrier effects nn nMOSFETs with high k dielectric.

As a first observation, the English of the manuscript is really poor. One can only guess what the authors intend to state and this makes evaluating the work that has been done quite challenging. Even the exact meaning of the title of the manuscript is not clear to me.

The devices that have been investigated are the 28nm high-k metal gate fabricated at UMC. These devices have been the subject of several research works by some of the authors of this paper. Besides those explicitly mentioned in the reference list (for instance ref 19 and 35) there are others that are not mentioned in this manuscript such as:

Chou, Ching-Chuan & Shen, Tien-Szu & Chen, Jian-Ming & Chang, Cheng-Hsun-Tony & Wang, Shea-Jue & Lan, Wen-How & Wang, Mu-Chun. (2020). Uniformity of Gate Dielectric for I/O and Core HK/MG pMOSFETs with Nitridation Treatments. Journal of Electronic Materials. 10.1007/s11664-020-08182-y.

Wang, Shea-Jue & Wang, Mu-Chun & Lee, Win-Der & Yang, Jie-Min & Huang, L.S. & Huang, Heng-Sheng. (2014). Gate Leakage for 28 nm Stacked HfZrOx Dielectric of p-Channel MOSFETs After Decoupled Plasma Nitridation Treatment With Annealing Temperatures. IEEE Transactions on Plasma Science. 42. 3712 - 3715. 10.1109/TPS.2014.2349005.

The reason I mention this is becasue in all the above four publications the process flow is reported in detail. In particular, Fig. 3 and Fig. 4 in the present manuscript are reported (with some irrelevant graphic changes, such as colors and shapes of the process flow chart) in several other papers. There is therefore no need to repeat it verbatim in yet another paper: a clear reference to any one of the previously published works would be sufficient.

Besides the poor English, that makes understanding the content of the introduction quite difficult, it can be safely assumed that the issue of hot carrier in MOS devices is a very well known subject and such a long discussion can be safely omitted (including Fig. 1 and Fig. 2) with a reference to one of the many comprehensive reviews dealing with the subject.

As far as the model for the interpretation of the results is concerned, the authors just assume a quite simple model (ref. 5, 1984). While I too believe that that work has been important in this field, hot carrier degradation is a much more complex subject and the issue of the model to be used for predicting lifetime is not at all straightforward. See for instance:

Mahapatra and U. Sharma, "A Review of Hot Carrier Degradation in n-Channel MOSFETs—Part I: Physical Mechanism," in IEEE Transactions on Electron Devices, vol. 67, no. 7, pp. 2660-2671, July 2020, doi: 10.1109/TED.2020.2994302.

Mahapatra and U. Sharma, "A Review of Hot Carrier Degradation in n-Channel MOSFETs—Part II: Technology Scaling," in IEEE Transactions on Electron Devices, vol. 67, no. 7, pp. 2672-2681, July 2020, doi: 10.1109/TED.2020.2994301.

Without a careful discussion of the applicability of a model, and with the problems in the English, the impression one gets from the paper is that the authors have performed some measurements (this is certainly interesting) and that have searched for the first model they could find to try to make sense of the results. Please understand me: I am not saying that this is what was done, I am saying that if this is not the case, it does not appear clear form the manuscript.

About the data presentation, I wonder why the authors have made the choice of plotting the drift in the threshold voltage vs the time on a linear scale, while plotting the same quantity vs time on a log scale could have provided a better idea of the type of dependence on time (see for instance the review I mentioned above).

Finally, there should be at least consistency in the names of the involved quantities. Unless I am mistaken, sometimes IDS is used and some other times IS is used for the same quantity? This adds to the confusion and really should be avoided.

I do not see how this work can be revised to reach publication standards at this stage. However, if the authors feel that they do have significant experimental results that are worth of publication, they should really make an effort to organize their material in a concise and consistent form and try to do a much better job in explaining what they have been done and why they believe that they have the right model to make sense of their results.

Author Response

The paper deals with the investigation of hot carrier effects for nMOSFETs with high k dielectric.

  1. As a first observation, the English of the manuscript is really poor. One can only guess what the authors intend to state and this makes evaluating the work that has been done quite challenging. Even the exact meaning of the title of the manuscript is not clear to me.
  2. The devices that have been investigated are the 28nm high-k metal gate fabricated at UMC. These devices have been the subject of several research works by some of the authors of this paper. Besides those explicitly mentioned in the reference list (for instance ref 19 and 35) there are others that are not mentioned in this manuscript such as:

Chou, Ching-Chuan & Shen, Tien-Szu & Chen, Jian-Ming & Chang,  Cheng-Hsun-Tony & Wang, Shea-Jue & Lan, Wen-How & Wang, Mu-Chun. (2020). Uniformity of Gate Dielectric for I/O and Core HK/MG pMOSFETs with Nitridation Treatments. Journal of Electronic Materials. 10.1007/s11664-020-08182-y.

Wang, Shea-Jue & Wang, Mu-Chun & Lee, Win-Der & Yang, Jie-Min & Huang, L.S. & Huang, Heng-Sheng. (2014). Gate Leakage for 28 nm Stacked HfZrOx Dielectric of p-Channel MOSFETs After Decoupled Plasma Nitridation Treatment With Annealing Temperatures. IEEE Transactions on Plasma Science. 42. 3712 - 3715. 10.1109/TPS.2014.2349005.

  1. The reason I mention this is because in all the above four publications the process flow is reported in detail. In particular, Fig. 3 and Fig. 4 in the present manuscript are reported (with some irrelevant graphic changes, such as colors and shapes of the process flow chart) in several other papers. There is therefore no need to repeat it verbatim in yet another paper: a clear reference to any one of the previously published works would be sufficient.
  2. Besides the poor English, that makes understanding the content of the introduction quite difficult, it can be safely assumed that the issue of hot carrier in MOS devices is a very well known subject and such a long discussion can be safely omitted (including Fig. 1 and Fig. 2) with a reference to one of the many comprehensive reviews dealing with the subject.
  3. As far as the model for the interpretation of the results is concerned, the authors just assume a quite simple model (ref. 5, 1984). While I too believe that that work has been important in this field, hot carrier degradation is a much more complex subject and the issue of the model to be used for predicting lifetime is not at all straightforward. See for instance: Mahapatra and U. Sharma, "A Review of Hot Carrier Degradation in n-Channel MOSFETs—Part I: Physical Mechanism," in IEEE Transactions on Electron Devices, vol. 67, no. 7, pp. 2660-2671, July 2020, doi: 10.1109/TED.2020.2994302. Mahapatra and U. Sharma, "A Review of Hot Carrier Degradation in n-Channel MOSFETs—Part II: Technology Scaling," in IEEE Transactions on Electron Devices, vol. 67, no. 7, pp. 2672-2681, July 2020, doi: 10.1109/TED.2020.2994301.
  4. Without a careful discussion of the applicability of a model, and with the problems in the English, the impression one gets from the paper is that the authors have performed some measurements (this is certainly interesting) and that have searched for the first model they could find to try to make sense of the results. Please understand me: I am not saying that this is what was done, I am saying that if this is not the case, it does not appear clear form the manuscript.
  5. About the data presentation, I wonder why the authors have made the choice of plotting the drift in the threshold voltage vs the time on a linear scale, while plotting the same quantity vs time on a log scale could have provided a better idea of the type of dependence on time (see for instance the review I mentioned above).
  6. Finally, there should be at least consistency in the names of the involved quantities. Unless I am mistaken, sometimes IDS is used and some other times IS is used for the same quantity? This adds to the confusion and really should be avoided.

I do not see how this work can be revised to reach publication standards at this stage. However, if the authors feel that they do have significant experimental results that are worth of publication, they should really make an effort to organize their material in a concise and consistent form and try to do a much better job in explaining what they have been done and why they believe that they have the right model to make sense of their results.

Reply:

        First of all, we really appreciate that you give us the impressive advice and keep a close eye on the quality of this journal. Thanks a lot. Furthermore, we will follow your concerns, step by step, to update or illustrate them.

  1. For the English quality, we send this manuscript to the proofreading service. The service website is “www.editage.com”, which provides the native English proofreading. In this article, we found the trend shift of bulk current before and after HC stress. Furthermore, the decrease of bulk current after HC stress is not only the increase of VT indirectly causing the decrease of IDS and reducing the impact ionization rate, but the increase of recombination rate coming from the diffusion current, observed at the deep nano-node device, referring the updated Reference 36. Due to these additional remarks, we believe that this article is different from the published literature and readable and referable, especially for the high-k devices. Thanks for noticing us this insufficient description.
  2. In the published article, the hidden rule is to avoid the self-quotation too much. Basically, one article can quote the maximum reference number from their team is two. Thus, we didn’t put many published literature from this team or the cooperated team. However, because you show your kindness and advice, we add one more as Ref. 43 and refer an article from the cooperated team as Ref. 36.
  3. Your concern is fine. We remove the Fig. 4. If some reader is interesting in the process flow in detail, he(she) can refer the quoted reference 21. For the Fig. 3, we keep it because not all of readers have the sense about the growth of stacked high-k gate dielectric. Furthermore, this schematic profile can strengthen the whole picture between HC stress and device formation. Thanks for considering this part.
  4. Thanks for expecting us the improvement of English writing. After discussion and referring the other reviewers’ comments, we think the Figs. 1 and 2 are able to be kept valuably, which is convenient to the readers. This history statement can clearly illustrate the HC mechanisms for DAHC and CHC stress methods and the dominant stress from the DAHC to CHC method is changed around at the 180-nm node. We know this introduction part for HC experts as reviewers’ roles is common. However, it’s still beneficial to the readers, who want to enter this reliability area to gain the relationship between high-k process and HC stress.
  5. Thanks for your recommendation for two recent HC literature. We quote them and made a table to illustrate the difference among the former and present achievements in different process technologies or growth of gate dielectric, which is suggested by the Reviewer #2. As you mentioned the hot carrier degradation is a complex subject, we really agree that. In this work, we didn’t consider the HC degradation in dynamic mechanism. Thanks for giving us the recommended reference labeled as Reference 12 to talk about the ac and dc stress. In the past, the illustration HC stress in ac stress was used by a ring oscillator to check the frequency shift like VT shift. Different ring oscillators with inverter or NAND mode as well as considering the number of stages with odd stages or even stages will show the different performance of frequency shift. The relationship between the dc and ac stress is also the other research topic, including the enhancement of degradation in temperature stress. Here, we focus on the degradation of substrate current before and after HC stress, otherwise the proposed article will be defocused.
  6. Thanks for your illustration. After these measurement experiments and data analysis, the team has ever presented to the wafer providers. They recognized our great and useful contribution. Couple of the sensed parameters and the degradation mechanisms were adopted in the adjustment of process development to establish the optimal process flow. However, due to the commercial confidence, we can’t tell which one was quoted. In summary, this work was indeed done well and favored both at 28-nm node. We ask the wafer providing team for giving us some favor in article publication. Ultimately, they agreed that, but the detail process information can’t be addressed in the published literature. We know you hope all of accepted papers in electronics journal can fit the higher standard quality. Thus, we believe that you are a responsible reviewer. Thanks again.
  1. For the HC stress to the tested device, people can consider the power law (ΔVT= A (t_stress)n ) if the lifetime prediction of the tested devices is very concerned. However, in this work, the difference of VT shift during these HC stress methods and the decrease of substrate current before and after HC stress are more concerned in this article. Therefore, using the linear scale maybe is a better way to see the trend of VT shift under the high or low gate field as well as the various channel devices. In discussion, we add the view of capability of trap repair under different gate fields. This supplement can demonstrate the distribution trend of VT shift vs. stress time under the higher gate field for CHC stress is similar, but not under the lower field. However, this phenomenon for DAHC stress is not obvious due to the gate field lower.
  2. Thanks for your mention. IDS means the current from the drain site to source site in MOSFET. For the IS, it means the current is sensed at the source site only. According to the Kirchhoff current loop, the ID from drain site can be separated as three parts: ISUB, IG, and IS. For a normal MOSFET, the gate leakage is the smallest in four sub-currents. Thus, we temporarily ignore it in this experiment. Then, we consider the ratio of ISUB/IS. If this ratio is hugely increased, the impact ionization rate contributed to ISUB is increased more. This is why we investigate the relationship of ISUB/IS vs. VGS and ISUB vs. VGS. To avoid the label confusion in reading, we add the illustration of IS in this article clearly and check the whole article again.

Finally, no matter what the final decision is, we all really appreciate you for giving us the precious advice. If we need to transfer this article to the other journal, your comments really can help us more.

Round 2

Reviewer 2 Report

The authors have not satisfactorily addressed the previous concerns raised. In addition, I have the following comments in the revised manuscript.

The research gap is not clearly mentioned after the concise literature review. I will recommend to find out the research gap explaining what is the lack in the literature and how this paper is fulfilling the gap?

Table 1 cannot be considered as a summary of the literature. It appears to me that the authors just want to address the comments rather than improving the paper quality. I will recommend you to see some of the high-quality papers that summarise the findings.

There are many terminological issues, such as authors' team. I would recommend you to seek help from a technical English editor.

I will also recommend adding more literature survey. 

Try to understand the structure of a manuscript, especially the conclusion section. It is not in practice to include figures and put references in this section. I will recommend you to see the guidelines.

Round 3

Reviewer 2 Report

Accepted